# Permutation-based Causal Inference Algorithms with Interventions

**Yuhao Wang**
Laboratory for Information and Decision Systems
and Institute for Data, Systems and Society
Massachusetts Institute of Technology
Cambridge, MA 02139
yuhaow@mit.edu

**Liam Solus**
Department of Mathematics
KTH Royal Institute of Technology
Stockholm, Sweden
solus@kth.se

**Karren Dai Yang**
Institute for Data, Systems and Society
and Broad Institute of MIT and Harvard
Massachusetts Institute of Technology
Cambridge, MA 02139
karren@mit.edu

**Caroline Uhler**
Laboratory for Information and Decision Systems
and Institute for Data, Systems and Society
Massachusetts Institute of Technology
Cambridge, MA 02139
cuhler@mit.edu

## Abstract

Learning directed acyclic graphs using both observational and interventional data is now a fundamentally important problem due to recent technological developments in genomics that generate such single-cell gene expression data at a very large scale. In order to utilize this data for learning gene regulatory networks, efficient and reliable causal inference algorithms are needed that can make use of both observational and interventional data. In this paper, we present two algorithms of this type and prove that both are consistent under the faithfulness assumption. These algorithms are interventional adaptations of the Greedy SP algorithm and are the first algorithms using both observational and interventional data with consistency guarantees. Moreover, these algorithms have the advantage that they are nonparametric, which makes them useful also for analyzing non-Gaussian data. In this paper, we present these two algorithms and their consistency guarantees, and we analyze their performance on simulated data, protein signaling data, and single-cell gene expression data.

## 1 Introduction

Discovering causal relations is a fundamental problem across a wide variety of disciplines including computational biology, epidemiology, sociology, and economics [5, 18, 20, 22]. DAG models can be used to encode causal relations in terms of a directed acyclic graph (DAG) $\mathcal{G}$, where each node is associated to a random variable and the arrows represent their causal influences on one another. The non-arrows of $\mathcal{G}$ encode a collection of conditional independence (CI) relations through the so-called *Markov properties*. While DAG models are extraordinarily popular within the aforementioned research fields, it is in general a difficult task to recover the underlying DAG $\mathcal{G}$ from samples from the joint distribution on the nodes. In fact, since different DAGs can encode the same set of CI relations, from observational data alone the underlying DAG $\mathcal{G}$ is in general only identifiable up to *Markov equivalence*, and *interventional* data is needed to identify the complete DAG.

In recent years, the new *drop-seq* technology has allowed obtaining high-resolution observational single-cell gene expression data at a very large scale [12]. In addition, earlier this year this technology

was combined with the CRISPR/Cas9 system into *perturb-seq*, a technology that allows obtaining high-throughput interventional gene expression data [4]. An imminent question now is how to make use of a combination of observational and interventional data (of the order of 100,000 cells / samples on 20,000 genes / variables) in the causal discovery process. Therefore, the development of efficient and consistent algorithms using both observational and interventional data that are implementable within genomics is now a crucial goal. This is the purpose of the present paper.

The remainder of this paper is structured as follows: In Section 2 we discuss related work. Then in Section 3, we recall fundamental facts about DAG models and causal inference that we will use in the coming sections. In Section 4, we present the two algorithms and discuss their consistency guarantees. In Section 5, we analyze the performance of the two algorithms on both simulated and real datasets. We end with a short discussion in Section 6.

## 2   Related Work

Causal inference algorithms based on observational data can be classified into three categories: constraint-based, score-based, and hybrid methods. *Constraint-based methods*, such as the *PC algorithm* [22], treat causal inference as a constraint satisfaction problem and rely on CI tests to recover the model via its Markov properties. *Score-based methods*, on the other hand, assign a score function such as the Bayesian Information Criterion (BIC) to each DAG and optimize the score via greedy approaches. An example is the prominent *Greedy Equivalence Search (GES)* [14]. *Hybrid methods* either alternate between score-based and constraint-based updates, as in *Max-Min Hill-Climbing* [26], or use score functions based on CI tests, as in the recently introduced *Greedy SP* algorithm [23].

Based on the growing need for efficient and consistent algorithms that accommodate observational and interventional data [4], it is natural to consider extensions of the previously described algorithms that can accommodate interventional data. Such options have been considered in [8], in which the authors propose GIES, an extension of GES that accounts for interventional data. This algorithm can be viewed as a greedy approach to $\ell_0$-*penalized maximum likelihood estimation* with interventional data, an otherwise computationally infeasible score-based approach. Hence GIES is a parametric approach (relying on Gaussianity) and while it has been applied to real data [8, 9, 15], we will demonstrate via an example in Section 3 that it is in general not consistent. In this paper, we assume causal sufficiency, i.e., that there are no latent confounders in the data-generating DAG. In addition, we assume that the interventional targets are known. Methods such as ACI [13], HEJ [10], COmbINE [25] and ICP [15] allow for latent confounders with possibly unknown interventional targets. In addition, other methods have been developed specifically for the analysis of gene expression data [19]. A comparison of the method presented here and some of these methods in the context of gene expression data is given in the Supplementary Material.

The main purpose of this paper is to provide the first algorithms (apart from enumerating all DAGs) for causal inference based on observational and interventional data with consistency guarantees. These algorithms are adaptations of the Greedy SP algorithm [23]. As compared to GIES, another advantage of these algorithms is that they are nonparametric and hence do not assume Gaussianity, a feature that is crucial for applications to gene expression data which is inherently non-Gaussian.

## 3   Preliminaries

**DAG models.** Given a DAG $\mathcal{G} = ([p], A)$ with node set $[p] := \{1, \ldots, p\}$ and a collection of arrows $A$, we associate the nodes of $\mathcal{G}$ to a random vector $(X_1, \ldots, X_p)$ with joint probability distribution $\mathbb{P}$. For a subset of nodes $S \subset [p]$, we let $\mathrm{Pa}_{\mathcal{G}}(S)$, $\mathrm{An}_{\mathcal{G}}(S)$, $\mathrm{Ch}_{\mathcal{G}}(S)$, $\mathrm{De}_{\mathcal{G}}(S)$, and $\mathrm{Nd}_{\mathcal{G}}(S)$, denote the *parents, ancestors, children, descendants*, and *nondescendants* of $S$ in $\mathcal{G}$. Here, we use the typical graph theoretical definitions of these terms as given in [11]. By the Markov property, the collection of non-arrows of $\mathcal{G}$ encode a set of CI relations $X_i \perp\!\!\!\perp X_{\mathrm{Nd}(i) \setminus \mathrm{Pa}(i)} \mid X_{\mathrm{Pa}(i)}$. A distribution $\mathbb{P}$ is said to satisfy the *Markov assumption* (a.k.a. be *Markov*) with respect to $\mathcal{G}$ if it entails these CI relations. A fundamental result about DAG models is that the complete set of CI relations implied by the Markov assumption for $\mathcal{G}$ is given by the *d-separation* relations in $\mathcal{G}$ [11, Section 3.2.2]; i.e., $\mathbb{P}$ satisfies the Markov assumption with respect to $\mathcal{G}$ if and only if $X_A \perp\!\!\!\perp X_B \mid X_C$ in $\mathbb{P}$ whenever $A$ and $B$ are

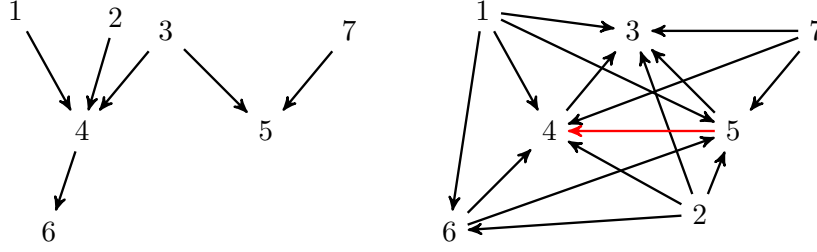

Figure 1: A generating DAG (left) and its GIES local maxima (right) for which GIES is not consistent.

$d$-separated in $\mathcal{G}$ given $C$. The *faithfulness assumption* is the assertion that the only CI relations entailed by $\mathbb{P}$ are those implied by $d$-separation in $\mathcal{G}$.

Two DAGs $\mathcal{G}$ and $\mathcal{H}$ with the same set of $d$-separation statements are called *Markov equivalent*, and the complete set of DAGs that are Markov equivalent to $\mathcal{G}$ is called its *Markov equivalence class* (MEC), denoted $[\mathcal{G}]$. The MEC of $\mathcal{G}$ is represented combinatorially by a partially directed graph $\widehat{\mathcal{G}} := ([p], D, E)$, called its *CP-DAG* or *essential graph* [1]. The arrows $D$ are precisely those arrows in $\mathcal{G}$ that have the same orientation in all members of $[\mathcal{G}]$, and the edges $E$ represent those arrows that change direction between distinct members of the MEC. In [2], the authors give a transformational characterization of the members of $[\mathcal{G}]$. An arrow $i \rightarrow j$ in $\mathcal{G}$ is called a *covered arrow* if $\mathrm{Pa}_{\mathcal{G}}(j) = \mathrm{Pa}_{\mathcal{G}}(i) \cup \{i\}$. Two DAGs $\mathcal{G}$ and $\mathcal{H}$ are Markov equivalent if and only if there exists a sequence of covered arrow reversals transforming $\mathcal{G}$ into $\mathcal{H}$ [2]. This transformational characterization plays a fundamental role in GES [14], GIES [8], Greedy SP [23], as well as the algorithms we introduce in this paper.

**Learning from Interventions.** In this paper, we consider multiple *interventions*. Given an ordered list of subsets of $[p]$ denoted by $\mathcal{I} := \{I_1, I_2, \ldots, I_K\}$, for each $I_j$ we generate an *interventional distribution*, denoted $\mathbb{P}^j$, by forcing the random variables $X_i$ for $i \in I_j$ to the value of some independent random variables. We assume throughout that $I_j = \emptyset$ for some $j$, i.e., that we have access to a combination of observational and interventional data. If $\mathbb{P}$ is Markov with respect to $\mathcal{G} = ([p], A)$, then the *intervention DAG of $I_j$* is the subDAG $\mathcal{G}^j := ([p], A^j)$ where $A^j = \{(i, j) \in A : j \notin I_j\}$; i.e., $\mathcal{G}^j$ is given by removing the incoming arrows to all intervened nodes in $\mathcal{G}$. Notice that $\mathbb{P}^j$ is always Markov with respect to $\mathcal{G}^j$. This fact allows us to naturally extend the notions of Markov equivalence and essential graphs to the interventional setting, as described in [8]. Two DAGs $\mathcal{G}$ and $\mathcal{H}$ are $\mathcal{I}$-*Markov equivalent* for the collection of interventions $\mathcal{I}$ if they have the same skeleton and the same set of immoralities, and if $\mathcal{G}^j$ and $\mathcal{H}^j$ have the same skeleton for all $j = 1, \ldots, K$ [8, Theorem 10]. Hence, any two $\mathcal{I}$-Markov equivalent DAGs lie in the same MEC. The $\mathcal{I}$-*Markov equivalence class* ($\mathcal{I}$-MEC) of $\mathcal{G}$ is denoted $[\mathcal{G}]_{\mathcal{I}}$. The $\mathcal{I}$-*essential graph* of $\mathcal{G}$ is the partially directed graph $\widehat{\mathcal{G}}_{\mathcal{I}} := \left([p], \cup_{j=1}^{K} D^j, \cup_{j=1}^{K} E^j\right)$, where $\widehat{\mathcal{G}}^j = ([p], D^j, E^j)$. The arrows of $\widehat{\mathcal{G}}_{\mathcal{I}}$ are called $\mathcal{I}$-*essential arrows* of $\mathcal{G}$.

**Greedy Interventional Equivalence Search (GIES).** GIES is a three-phase score-based algorithm: In the *forward phase*, GIES initializes with an empty $\mathcal{I}$-essential graph $\widehat{\mathcal{G}}_0$. Then it sequentially steps from one $\mathcal{I}$-essential graph $\widehat{\mathcal{G}}_i$ to a larger one $\widehat{\mathcal{G}}_{i+1}$ given by adding a single arrow to $\widehat{\mathcal{G}}_i$. In the *backward phase*, it steps from one essential graph $\widehat{\mathcal{G}}_i$ to a smaller one $\widehat{\mathcal{G}}_{i+1}$ containing precisely one less arrow than $\widehat{\mathcal{G}}_i$. In the *turning phase*, the algorithm reverses the direction of arrows. It first considers reversals of non-$\mathcal{I}$-essential arrows and then the reversal of $\mathcal{I}$-essential arrows, allowing it to move between $\mathcal{I}$-MECs. At each step in all phases the maximal scoring candidate is chosen, and the phase is only terminated when no higher-scoring $\mathcal{I}$-essential graph exists. GIES repeatedly executes the forward, backward, and turning phases, in that order, until no higher-scoring $\mathcal{I}$-essential graph can be found. It is amenable to any score that is constant on an $I$-MEC, such as the BIC.

The question whether GIES is consistent, was left open in [8]. We now prove that GIES is in general not consistent; i.e., if $n_j$ i.i.d. samples are drawn from the interventional distribution $\mathbb{P}^j$, then even as $n_1 + \cdots + n_K \rightarrow \infty$ and under the faithfulness assumption, GIES may not recover the optimal $\mathcal{I}$-MEC with probability 1. Consider the data-generating DAG depicted on the left in Figure 1.

---

**Algorithm 1:**

---

**Input:** Observations $\hat{X}$, an initial permutation $\pi_0$, a threshold $\delta_n > \sum_{k=1}^K \lambda_{n_k}$, and a set of interventional targets $\mathcal{I} = \{I_1, \ldots, I_K\}$.

**Output:** A permutation $\pi$ and its minimal I-MAP $\mathcal{G}_\pi$.

**1** Set $\mathcal{G}_\pi := \underset{\mathcal{G} \text{ consistent with } \pi}{\mathrm{argmax}} \; \mathrm{Score}(\mathcal{G})$;

**2** Using a depth-first search approach with root $\pi$, search for a permutation $\pi_s$ with $\mathrm{Score}(\mathcal{G}_{\pi_s}) > \mathrm{Score}(\mathcal{G}_\pi)$ that is connected to $\pi$ through a sequence of permutations

$$\pi_0 = \pi, \pi_1, \cdots, \pi_{s-1}, \pi_s,$$

where each permutation $\pi_k$ is produced from $\pi_{k-1}$ by a transposition that corresponds to a covered edge in $\mathcal{G}_{\pi_{k-1}}$ such that $\mathrm{Score}(\mathcal{G}_{\pi_k}) > \mathrm{Score}(\mathcal{G}_{\pi_{k-1}}) - \delta_n$. If no such $\mathcal{G}_{\pi_s}$ exists, return $\pi$ and $\mathcal{G}_\pi$; else set $\pi := \pi_s$ and repeat.

---

Suppose we take interventions $\mathcal{I}$ consisting of $I_1 = \emptyset, I_2 = \{4\}, I_3 = \{5\}$, and that GIES arrives at the DAG $\mathcal{G}$ depicted on the right in Figure 1. If the data collected grows as $n_1 = Cn_2 = Cn_3$ for some constant $C > 1$, then we can show that the BIC score of $\mathcal{G}$ is a local maximum with probability $\frac{1}{2}$ as $n_1$ tends to infinity. The proof of this fact relies on the observation that GIES must initialize the turning phase at $\mathcal{G}$, and that $\mathcal{G}$ contains precisely one covered arrow $5 \to 4$, which is colored red in Figure 1. The full proof is given in the Supplementary Material.

**Greedy SP.** In this paper we adapt the hybrid algorithm Greedy SP to provide consistent algorithms that use both interventional and observational data. Greedy SP is a permutation-based algorithm that associates a DAG to every permutation of the random variables and greedily updates the DAG by transposing elements of the permutation. More precisely, given a set of observed CI relations $\mathcal{C}$ and a permutation $\pi = \pi_1 \cdots \pi_p$, the Greedy SP algorithm assigns a DAG $\mathcal{G}_\pi := ([p], A_\pi)$ to $\pi$ via the rule

$$\pi_i \to \pi_j \in A_\pi \quad \Longleftrightarrow \quad i < j \text{ and } \pi_i \not\perp\!\!\!\perp \pi_j \mid \{\pi_1, \ldots, \pi_{\max(i,j)}\} \backslash \{\pi_i, \pi_j\},$$

for all $1 \leq i < j \leq p$. The DAG $\mathcal{G}_\pi$ is a *minimal I-MAP* (independence map) with respect to $\mathcal{C}$, since any DAG $\mathcal{G}_\pi$ is Markov with respect to $\mathcal{C}$ and any proper subDAG of $\mathcal{G}_\pi$ encodes a CI relation that is not in $\mathcal{C}$ [17]. Using a depth-first search approach, the algorithm reverses covered edges in $\mathcal{G}_\pi$, takes a linear extension $\tau$ of the resulting DAG and re-evaluates against $\mathcal{C}$ to see if $\mathcal{G}_\tau$ has fewer arrows than $\mathcal{G}_\pi$. If so, the algorithm reinitializes at $\tau$, and repeats this process until no sparser DAG can be recovered. In the observational setting, Greedy SP is known to be consistent whenever the data-generating distribution is faithful to the sparsest DAG [23].

## 4 Two Permutation-Based Algorithms with Interventions

We now introduce our two interventional adaptations of Greedy SP and prove that they are consistent under the faithfulness assumption. In the first algorithm, presented in Algorithm 1, we use the same moves as Greedy SP, but we optimize with respect to a new score function that utilizes interventional data, namely the sum of the interventional BIC scores. To be more precise, for a collection of interventions $\mathcal{I} = \{I_1, \ldots, I_K\}$, the new score function is

$$\mathrm{Score}(\mathcal{G}) := \sum_{k=1}^K \left( \underset{(A,\Omega) \in \mathcal{G}^k}{\mathrm{maximize}} \; \ell_k \left( \hat{X}^k; A, \Omega \right) \right) - \sum_{k=1}^K \lambda_{n_k} |\mathcal{G}^k|,$$

where $\ell_k$ denotes the log-likelihood of the interventional distribution $\mathbb{P}^k$, $(A, \Omega)$ are any parameters consistent with $\mathcal{G}^k$, $|\mathcal{G}|$ denotes the number of arrows in $\mathcal{G}$, and $\lambda_{n_k} = \frac{\log n_k}{n_k}$.

When Algorithm 1 has access to observational and interventional data, then uniform consistency follows using similar techniques to those used to prove uniform consistency of Greedy SP in [23]. A full proof of the following consistency result for Algorithm 1 is given in the Supplementary Material.

**Theorem 4.1.** *Suppose $\mathbb{P}$ is Markov with respect to an unknown I-MAP $\mathcal{G}_{\pi^*}$. Suppose also that observational and interventional data are drawn from $\mathbb{P}$ for a collection of interventional targets $\mathcal{I} = \{I_1 := \emptyset, I_2, \ldots, I_K\}$. If $\mathbb{P}^k$ is faithful to $(\mathcal{G}_{\pi^*})^k$ for all $k \in [K]$, then Algorithm 1 returns the $\mathcal{I}$-MEC of the data-generating DAG $\mathcal{G}_{\pi^*}$ almost surely as $n_k \to \infty$ for all $k \in [K]$.*

---

**Algorithm 2:** Interventional Greedy SP (IGSP)

---

**Input:** A collection of interventional targets $\mathcal{I} = \{\mathcal{I}_1, \ldots, \mathcal{I}_K\}$ and a starting permutation $\pi_0$.
**Output:** A permutation $\pi$ and its minimal I-MAP $\mathcal{G}_\pi$.

---
1 Set $\mathcal{G} := \mathcal{G}_{\pi_0}$;
2 Using a depth-first-search approach with root $\pi$, search for a minimal I-MAP $\mathcal{G}_\tau$ with $|\mathcal{G}| > |\mathcal{G}_\tau|$ that is connected to $\mathcal{G}$ by a list of $\mathcal{I}$-covered edge reversals. Along the search, prioritize the $\mathcal{I}$-covered edges that are also $\mathcal{I}$-contradicting edges. If such $\mathcal{G}_\tau$ exists, set $\mathcal{G} := \mathcal{G}_\tau$, update the number of $\mathcal{I}$-contradicting edges, and repeat this step. If not, output $G_\tau$ with $|\mathcal{G}| = |\mathcal{G}_\tau|$ that is connected to $\mathcal{G}$ by a list of $\mathcal{I}$-covered edges and minimizes the number of $\mathcal{I}$-contradicting edges.

---

A problematic feature of Algorithm 1 from a computational perspective is the the slack parameter $\delta_n$. In fact, if this parameter were not included, then Algorithm 1 would not be consistent. This can be seen via an application of Algorithm 1 to the example depicted in Figure 1. Using the same set-up as the inconsistency example for GIES, suppose that the left-most DAG $\mathcal{G}$ in Figure 1 is the data generating DAG, and that we draw $n_k$ i.i.d. samples from the interventional distribution $\mathbb{P}^k$ for the collection of targets $\mathcal{I} = \{\mathcal{I}_1 = \emptyset, \mathcal{I}_2 = \{4\}, \mathcal{I}_3 = \{5\}\}$. Suppose also that $n_1 = Cn_2 = Cn_3$ for some constant $C > 1$, and now additionally assume that we initialize Algorithm 1 at the permutation $\pi = 1276543$. Then the minimal I-MAP $\mathcal{G}_\pi$ is precisely the DAG presented on the right in Figure 1. This DAG contains one covered arrow, namely $5 \rightarrow 4$. Reversing it produces the minimal I-MAP $\mathcal{G}_\tau$ for $\tau = 1276453$. Computing the score difference $\text{Score}(\mathcal{G}_\tau) - \text{Score}(\mathcal{G}_\pi)$ using [16, Lemma 5.1] shows that as $n_1$ tends to infinity, $\text{Score}(\mathcal{G}_\tau) < \text{Score}(\mathcal{G}_\pi)$ with probability $\frac{1}{2}$. Hence, Algorithm 1 would not be consistent without the slack parameter $\delta_n$. This calculation can be found in the Supplementary Material.

Our second interventional adaptation of the Greedy SP algorithm, presented in Algorithm 2, leaves the score function the same (i.e., the number of edges of the minimal I-MAP), but restricts the possible covered arrow reversals that can be queried at each step. In order to describe this restricted set of moves we provide the following definitions.

**Definition 4.2.** Let $\mathcal{I} = \{I_1, \ldots, I_K\}$ be a collection of interventions, and for $i, j \in [p]$ define the collection of indices
$$\mathcal{I}_{i \setminus j} := \{k \in [K] : i \in I_k \text{ and } j \notin I_k\}.$$
For a minimal I-MAP $\mathcal{G}_\pi$ we say that a covered arrow $i \rightarrow j \in \mathcal{G}_\pi$ is $\mathcal{I}$-*covered* if
$$\mathcal{I}_{i \setminus j} = \emptyset \quad \text{or} \quad i \rightarrow j \notin (\mathcal{G}^k)_\pi \quad \text{for all } k \in \mathcal{I}_{i \setminus j}.$$

**Definition 4.3.** We say that an arrow $i \rightarrow j \in \mathcal{G}_\pi$ is $\mathcal{I}$-*contradicting* if the following three conditions hold: (a) $\mathcal{I}_{i \setminus j} \cup \mathcal{I}_{j \setminus i} \neq \emptyset$, (b) $\mathcal{I}_{i \setminus j} = \emptyset$ or $i \perp\!\!\!\perp j$ in distribution $\mathbb{P}^k$ for all $k \in \mathcal{I}_{i \setminus j}$, (c) $\mathcal{I}_{j \setminus i} = \emptyset$ or there exists $k \in \mathcal{I}_{j \setminus i}$ such that $i \not\perp\!\!\!\perp j$ in distribution $\mathbb{P}^k$.

In the observational setting, GES and Greedy SP utilize covered arrow reversals to transition between members of a single MEC as well as between MECs [2, 3, 23]. Since an $\mathcal{I}$-MEC is characterized by the skeleta and immoralities of each of its interventional DAGs, $\mathcal{I}$-covered arrows represent the natural candidate for analogous transitionary moves between $\mathcal{I}$-MECs in the interventional setting. It is possible that reversing an $\mathcal{I}$-covered edge $i \rightarrow j$ in a minimal I-MAP $\mathcal{G}_\pi$ results in a new minimal I-MAP $\mathcal{G}_\tau$ that is in the same $\mathcal{I}$-MEC as $\mathcal{G}_\pi$. Namely, this happens when $i \rightarrow j$ is a non-$\mathcal{I}$-essential edge in $\mathcal{G}_\pi$. Similar to Greedy SP, Algorithm 2 implements a depth-first-search approach that allows for such $\mathcal{I}$-covered arrow reversals, but it prioritizes those $\mathcal{I}$-covered arrow reversals that produce a minimal I-MAP $\mathcal{G}_\tau$ that is not $\mathcal{I}$-Markov equivalent to $\mathcal{G}_\pi$. These arrows are $\mathcal{I}$-contradicting arrows. The result of this refined search via $\mathcal{I}$-covered arrow reversal is an algorithm that is consistent under the faithfulness assumption.

**Theorem 4.4.** *Algorithm 2 is consistent under the faithfulness assumption.*

The proof of Theorem 4.4 is given in the Supplementary Material. When only observational data is available, Algorithm 2 boils down to greedy SP. We remark that the number of queries conducted in a given step of Algorithm 2 is, in general, strictly less than in the purely observational setting. That is to say, $\mathcal{I}$-covered arrows generally constitute a strict subset of the covered arrows in a DAG. At first

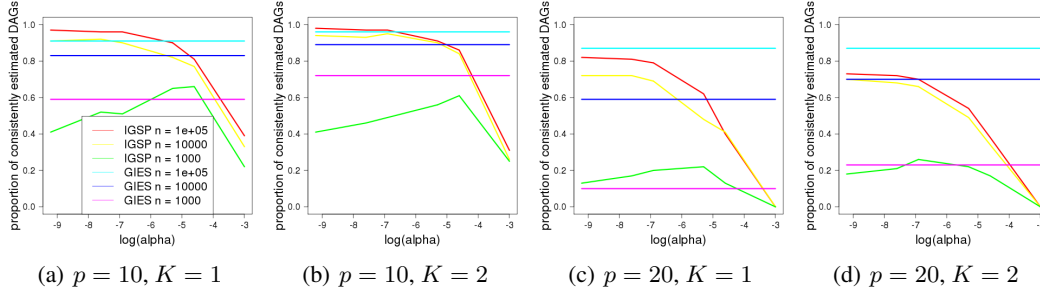

| (a) $p = 10, K = 1$ | (b) $p = 10, K = 2$ | (c) $p = 20, K = 1$ | (d) $p = 20, K = 2$ |

Figure 2: The proportion of consistently estimated DAGs for 100 Gaussian DAG models on $p$ nodes with $K$ single-node interventions.

glance, keeping track of the $\mathcal{I}$-covered edges may appear computationally inefficient. However, at each step we only need to update this list locally; so the computational complexity of the algorithm is not drastically impacted by this procedure. Hence, access to interventional data is beneficial in two ways: it allows to reduce the search directions at every step and it often allows to estimate the true DAG more accurately, since an $\mathcal{I}$-MEC is in general smaller than an MEC. Note that in this paper all the theoretical analysis are based on the low-dimensional setting, where $p \ll n$. The high-dimensional consistency of greedy SP is shown in [23], and it is not difficult to see that the same high-dimensional consistency guarantees also apply to IGSP.

## 5 Evaluation

In this section, we compare Algorithm 2, which we call *Interventional Greedy SP* (*IGSP*) with GIES on both simulated and real data. Algorithm 1 is of interest from a theoretical perspective, but it is computationally inefficient since it requires performing two variable selection procedures per update. Therefore, it will not be analyzed in this section. The code utilized for the following experiments can be found at `https://github.com/yuhaow/sp-intervention`.

### 5.1 Simulations

Our simulations are conducted for linear structural equation models with Gaussian noise:

$$(X_1, \ldots, X_p)^T = ((X_1, \ldots, X_p)A)^T + \epsilon,$$

where $\epsilon \sim \mathcal{N}(0, \mathbf{1}_p)$ and $A = (a_{ij})_{i,j=1}^p$ is an upper-triangular matrix of edge weights with $a_{ij} \neq 0$ if and only if $i \to j$ is an arrow in the underlying DAG $\mathcal{G}^*$. For each simulation study we generated 100 realizations of an (Erdös-Renyi) random $p$-node Gaussian DAG model for $p \in \{10, 20\}$ with an expected edge density of 1.5. The collections of interventional targets $\mathcal{I} = \{I_0 := \emptyset, I_1, \ldots, I_K\}$ always consist of the empty set $I_0$ together with $K = 1$ or 2. For $p = 10$, the size of each intervention set was 5 for $K = 1$ and 4 for $K = 2$. For $p = 20$, the size was increased up to 10 and 8 to keep the proportion of intervened nodes constant. In each study, we compared GIES with Algorithm 2 for $n$ samples for each intervention with $n = 10^3, 10^4, 10^5$. Figure 2 shows the proportion of consistently estimated DAGs as distributed by choice of cut-off parameter for partial correlation tests. Interestingly, although GIES is not consistent on random DAGs, in some cases it performs better than IGSP, in particular for smaller sample sizes. However, as implied by the consistency guarantees given in Theorem 4.4, IGSP performs better as the sample size increases.

We also conducted a focused simulation study on models for which the data-generating DAG $\mathcal{G}$ is that depicted on the left in Figure 1, for which GIES is not consistent. In this simulation study, we took 100 realizations of Gaussian models for the data-generating DAG $\mathcal{G}$ for which the nonzero edge-weights $a_{ij}$ were randomly drawn from $[-1, -c, ) \cup (c, 1]$ for $c = 0.1, 0.25, 0.5$. The interventional targets were $\mathcal{I} = \{I_0 = \emptyset, I_1\}$, where $I_1$ was uniformly at random chosen to be $\{4\}, \{5\}, \{4, 5\}$. Figure 3 shows, for each choice of $c$, the proportion of times $\mathcal{G}$ was consistently estimated as distributed by the choice of cut-off parameter for the partial correlation tests. We see from these plots that as expected from our theoretical results GIES recovers $\mathcal{G}$ at a lower rate than Algorithm 2.

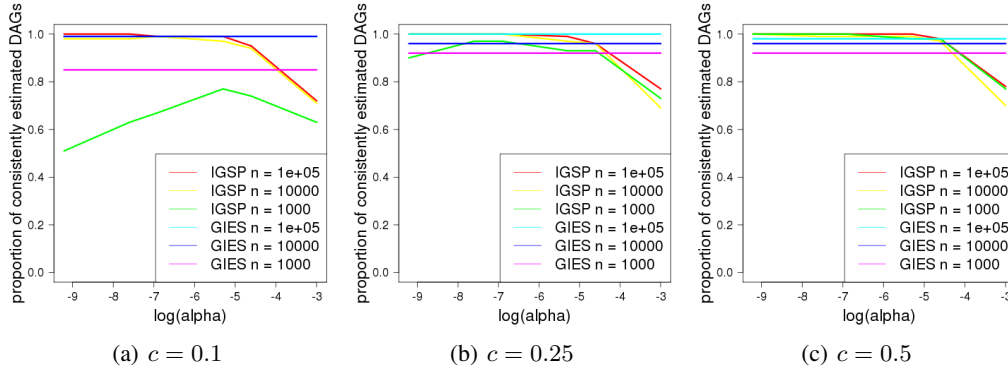

(a) $c = 0.1$          (b) $c = 0.25$          (c) $c = 0.5$

Figure 3: Proportion of times the DAG $\mathcal{G}$ from Figure 1 (left) is consistently estimated under GIES and Algorithm 2 for Gaussian DAG models with edge-weights drawn from $[-1, -c) \cup (c, 1]$.

## 5.2  Application to Real Data

In the following, we report results for studies conducted on two real datasets coming from genomics. The first dataset is the protein signaling dataset of *Sachs et al.* [21], and the second is the single-cell gene expression data generated using perturb-seq in [4].

**Analysis of protein signaling data.** The dataset of Sachs et al. [21] consists of 7466 measurements of the abundance of phosphoproteins and phospholipids recorded under different experimental conditions in primary human immune system cells. The different experimental conditions are generated using various reagents that inhibit or activate signaling nodes, and thereby correspond to interventions at different nodes in the protein signaling network. The dataset is purely interventional and most interventions take place at more than one target. Since some of the experimental perturbations effect receptor enzymes instead of the measured signaling molecules, we consider only the 5846 measurements in which the perturbations of receptor enzymes are identical. In this way, we can define the observational distribution to be that of molecule abundances in the model where only the receptor enzymes are perturbed. This results in $1755$ observational measurements and $4091$ interventional measurements. Table E.2 in the Supplementary Material summarizes the number of samples as well as the targets for each intervention. For this dataset we compared the GIES results reported in [9] with Algorithm 2 using both, a linear Gaussian and a kernel-based independence criterium [6, 24]. A crucial advantage of Algorithm 2 over GIES is that it is nonparametric and does not require Gaussianity. In particular, it supports kernel-based CI tests that are in general able to deal better with non-linear relationships and non-Gaussian noise, a feature that is typical of datasets such as this one.

For the GIES algorithm we present the results of [8] in which the authors varied the number of edge additions, deletions, and reversals as tuning parameters. For the linear Gaussian and kernel-based implementations of IGSP our tuning parameter is the cut-off value for the CI tests, just as in the simulated data studies in Section 5.1. Figure 4 reports our results for thirteen different cut-off values in $[10^{-4}, 0.7]$, which label the corresponding points in the plots. The linear Gaussian and kernel-based implementations of IGSP are comparable and generally both outperform GIES. The Supplementary Material contains a comparison of the results obtained by IGSP on this dataset to other recent methods that allow also for latent confounders, such as ACI, COmbINE and ICP.

**Analysis of perturb-seq gene expression data.** We analyzed the performance of GIES and IGSP on perturb-seq data published by Dixit et al. [4]. The dataset contains observational data as well as interventional data from ∼30,000 bone marrow-derived dendritic cells (BMDCs). Each data point contains gene expression measurements of 32,777 genes, and each interventional data point comes from a cell where a single gene has been targeted for deletion using the CRISPR/Cas9 system.

After processing the data for quality, the data consists of 992 observational samples and 13,435 interventional samples from eight gene deletions. The number of samples collected under each of the eight interventions is shown in the Supplementary Material. These interventions were chosen based

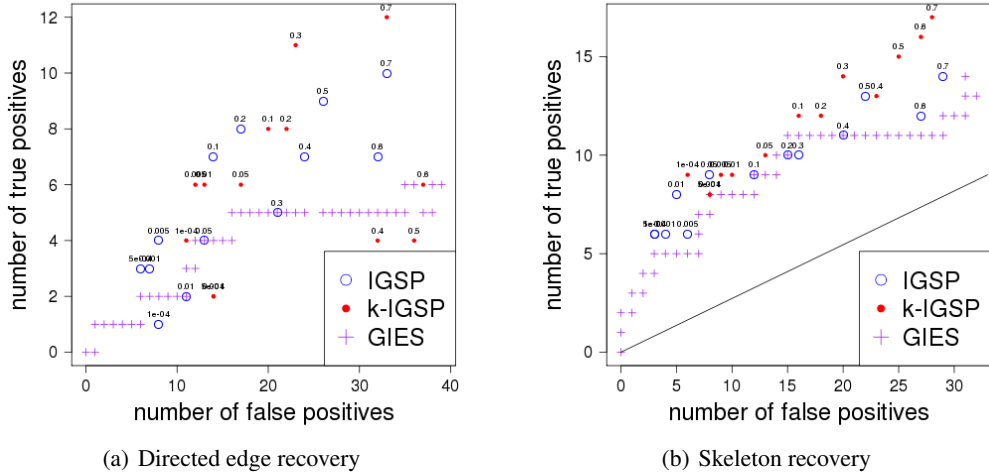

<table>
<tr><td>(a) Directed edge recovery</td><td>(b) Skeleton recovery</td></tr>
</table>

Figure 4: ROC plot of the models estimated from the data [21] using GIES as reported in [8] and the linear Gaussian and kernel-based versions of IGSP with different cut-off values for the CI tests. The solid line indicates the accuracy achieved by random guessing.

on empirical evidence that the gene deletion was effective[1]. We used GIES and IGSP to learn causal DAGs over 24 of the measured genes, including the ones targeted by the interventions, using both observational and interventional data. We followed [4] in focusing on these 24 genes, as they are general transcription factors known to regulate each other as well as numerous other genes [7].

We evaluated the learned causal DAGs based on their accuracy in predicting the true effects of each of the interventions (shown in Figure 5(a)) when leaving out the data for that intervention. Specifically, if the predicted DAG indicates an arrow from gene A to gene B, we count this as a true positive if knocking out gene A caused a significant change[2] in the distribution of gene B, and a false positive otherwise. For each inference algorithm and for every choice of the tuning parameters, we learned eight causal DAGs, each one trained with one of the interventional datasets being left out. We then evaluated each algorithm based on how well the causal DAGs are able to predict the corresponding held-out interventional data. As seen in Figure 5(b), IGSP predicted the held-out interventional data better than GIES (as implemented in the R-package pcalg) and random guessing, for a number of choices of the cut-off parameter. The true and reconstructed networks for both genomics datasets are shown in the Supplementary Material.

## 6 Discussion

We have presented two hybrid algorithms for causal inference using both observational and interventional data and we proved that both algorithms are consistent under the faithfulness assumption. These algorithms are both interventional adaptations of the Greedy SP algorithm and are the first algorithms of this type that have consistency guarantees. While Algorithm 1 suffers a high level of inefficiency, IGSP is implementable and competitive with the state-of-the-art, i.e., GIES. Moreover, IGSP has the distinct advantage that it is nonparametric and therefore does not require a linear Gaussian assumption on the data-generating distribution. We conducted real data studies for protein signaling and single-cell gene expression datasets, which are typically non-linear with non-Gaussian noise. In general, IGSP outperformed GIES. This purports IGSP as a viable method for analyzing the new high-resolution datasets now being produced by procedures such as perturb-seq. An important

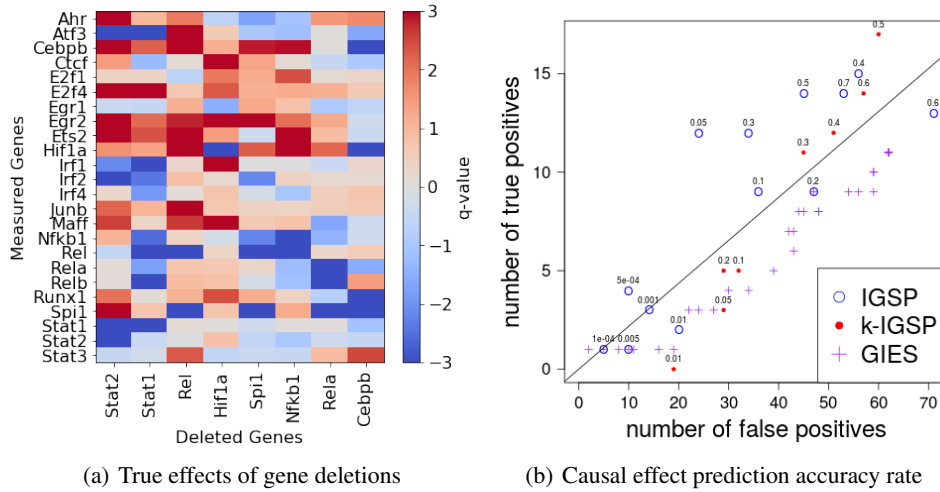

(a) True effects of gene deletions

(b) Causal effect prediction accuracy rate

Figure 5: (a) Heatmap of the true effects of each gene deletion on each measured gene. The q-value has the same magnitude as the log p-value of the Wilcoxon rank-sum test between the distributions of observational data and the interventional data. Positive and negative q-values indicate increased and decreased abundance as a result of deletion respectively. (b) ROC plot of prediction accuracy by the causal DAGs learned by IGSP and GIES. The solid line indicates the accuracy achieved by random guessing.

challenge for future work is to make these algorithms scale to 20,000 nodes, i.e., the typical number of genes in such studies. In addition, in future work it would be interesting to extend IGSP to allow for latent confounders. An advantage of not allowing for latent confounders is that a DAG is usually more identifiable. For example, if we consider a DAG with two observable nodes, a DAG without confounders is fully identifiable by intervening on only one of the two nodes, but the same is not true for a DAG with confounders.

## Acknowledgements

Yuhao Wang was supported by DARPA (W911NF-16-1-0551) and ONR (N00014-17-1-2147). Liam Solus was supported by an NSF Mathematical Sciences Postdoctoral Research Fellowship (DMS - 1606407). Karren Yang was supported by the MIT Department of Biological Engineering. Caroline Uhler was partially supported by DARPA (W911NF-16-1-0551), NSF (1651995) and ONR (N00014-17-1-2147). We thank Dr. Sofia Triantafillou from the University of Crete for helping us run COmbINE.

## Footnotes

[1]An intervention was considered effective if the distribution of the gene expression levels of the deleted gene is significantly different from the distribution of its expression levels without intervention, based on a Wilcoxon Rank-Sum test with $\alpha = 0.05$. Ineffective interventions on a gene are typically due to poor targeting ability of the guide-RNA designed for that gene.

[2]Based on a Wilcoxon Rank-Sum test with $\alpha = 0.05$, which is approximately equivalent to a q-value of magnitude $\geq 3$ in Figure 5(a)

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
