[Supplementary Material · nips_2017_appendix_2017_10_21.pdf]

# Supplementary Material: Permutation-based Causal Inference Algorithms with Interventions

**Yuhao Wang**
Laboratory for Information and Decision Systems
and Institute for Data, Systems and Society
Massachusetts Institute of Technology
Cambridge, MA 02139
yuhaow@mit.edu

**Liam Solus**
Department of Mathematics
KTH Royal Institute of Technology
Stockholm, Sweden
solus@kth.se

**Karren Dai Yang**
Institute for Data, Systems and Society
and Broad Institute of MIT and Harvard
Massachusetts Institute of Technology
Cambridge, MA 02139
karren@mit.edu

**Caroline Uhler**
Laboratory for Information and Decision Systems
and Institute for Data, Systems and Society
Massachusetts Institute of Technology
Cambridge, MA 02139
cuhler@mit.edu

## A  Counterexample to Consistency of GIES.

In the following, we verify that the example of GIES described in Section 3 is in fact a counterexample to consistency of GIES with the BIC score function. Recall that the DAG on the left in Figure 1, which we denote $\mathcal{G}_0$, is taken to be the data-generating DAG, and our collection of interventions is $\mathcal{I} = \{I_1 = \emptyset, I_2 = \{4\}, I_3 = \{5\}\}$. Suppose that the number of samples, $n_1, n_2, n_3$, drawn from the interventional distributions $\mathbb{P}^1, \mathbb{P}^2, \mathbb{P}^3$, satisfy $n_1 = Cn_2 = Cn_3$ for some constant $C > 1$, and that GIES arrives at the DAG $\mathcal{G}$ depicted on the right in Figure 1. Here, we also assume that the observational distribution $\mathbb{P}^1$ is faithful to $\mathcal{G}_0$. We claim that this DAG is a local maximum of the GIES algorithm.

To see this, first notice that since $5 \to 4$ is the only covered edge in $\mathcal{G}$, then its $\mathcal{I}$-MEC has size one. Also notice that the DAG $\mathcal{G}$ is the minimal I-MAP of $\mathcal{G}_\pi$ for the permutation $\pi = 1276543$. Therefore, by consistency of GES under faithfulness [1], deleting any edge of $\mathcal{G}$ would result in a DAG with a strictly higher BIC. Thus, it only remains to verify that $\mathcal{G}$ is a local maximum with respect to the turning phase. We begin by checking that turning the only covered arrow in $\mathcal{G}$ does not increase the BIC score function with probablilty 1. In the following, for a node $j$, we let $\mathcal{I}_{-j} := \mathcal{I}\setminus\{k \mid j \in I_k\}$. We may then express the score of $\mathcal{G}$ as

$$\text{Score}(\mathcal{G}, \hat{X}) := \sum_{j=1}^{p} s(j, \text{Pa}_j(\mathcal{G}), \hat{X}^{\mathcal{I}_{-j}}) + C - \lambda_n|\mathcal{G}|,$$

where $s(j, \text{Pa}_j(\mathcal{G}), \hat{X}^{\mathcal{I}_{-j}})$ is the log of the regression residual when regressing $j$ on $\text{Pa}_j(\mathcal{G})$ using the data from the truncated intervention set $\mathcal{I}_{-j}$. Formally,

$$s(j, \text{Pa}_j(\mathcal{G}), \hat{X}^{\mathcal{I}_{-j}}) = -\frac{1}{2}\frac{n_{-j}}{n}\log\left(\min_{a \in \mathbb{R}^{|\text{Pa}_j(\mathcal{G})|}} \sum_{k \in \mathcal{I}_{-j}} \|\hat{X}_j^k - \hat{X}_{\text{Pa}_j(\mathcal{G})}^k \cdot a\|_2^2/n_{-j}\right).$$

Now let $\mathcal{G}'$ denote the DAG produced by reversing the arrow $d \to c$ in $\mathcal{G}$, and let $\hat{\rho}_{i,j|S}^{\mathcal{I}}$ denote the partial correlation testing coefficient of $i$ and $j$ given some $S \subset [p]$ using interventional data

$\hat{X}^k, \forall k \in \mathcal{I}$. If we take $S = \mathrm{Pa}_{\mathcal{G}}(i) \cap \mathrm{Pa}_{\mathcal{G}}(j)$, then by [5, Lemma 5.1] we have that

$$\mathrm{Score}(\mathcal{G}', \hat{X}) - \mathrm{Score}(\mathcal{G}, \hat{X}) = s(c, S, \hat{X}^{\mathcal{I}-c}) - s(c, S \cup \{d\}, \hat{X}^{\mathcal{I}-c})$$
$$+ s(d, S \cup \{c\}, \hat{X}^{\mathcal{I}-d}) - s(d, S, \hat{X}^{\mathcal{I}-d}),$$
$$= -\frac{1}{2}\frac{n_{-d}}{n}\log(1 - (\hat{\rho}_{c,d|S}^{\mathcal{I}-d})^2) + \frac{1}{2}\frac{n_{-c}}{n}\log(1 - (\hat{\rho}_{c,d|S}^{\mathcal{I}-c})^2).$$

Since $n_1 = Cn_2 = Cn_3$ it follows that the distributions of $\hat{\rho}_{c,d|S}^{\mathcal{I}-d}$ and $\hat{\rho}_{c,d|S}^{\mathcal{I}-c}$ are always identical. Therefore, $\mathrm{Score}(\mathcal{G}', \hat{X}) < \mathrm{Score}(\mathcal{G}, \hat{X})$ with probability $\frac{1}{2}$.

It now only remains to verify that turning any non-covered edge in $\mathcal{G}$ increases the value of the BIC score function. Suppose that $\mathcal{G}'$ is a DAG produced from turning some edge in $\mathcal{G}$ other than $5 \to 4$. Since such an edge is not covered, $\mathcal{G}'$ will not be an independence map of the un-intervened distribution $\mathbb{P}^1$. Therefore, there exists some sufficiently large $C > 1$ such that the score of $\mathcal{G}$ is larger than $\mathcal{G}'$. This is because the score function for $C$ large enough is dominated by the part that depends on the observational data. Thus, we conclude that $\mathcal{G}$ is a local maximum of GIES. $\qquad \square$

## B  Counterexample to Consistency of Algorithm 1 without the Slack Factor

We now verify that the example described in Section 4 shows that Algorithm 1 without the use of the slack factor $\delta_n$ is not consistent. The proof of this statement is similar to that of the counterexample to consistency of GIES, and so we adopt the exact same set-up and notation used in the previous proof. Unlike GIES, Algorithm 1 only uses moves corresponding to reversals of covered edges in the observational DAG $\mathcal{G}_0$, depicted on the right in Figure 1. Thus, the only possible move Algorithm 1 can make is to reverse the covered arrow $5 \to 4$. If we denote the resulting graph by $\mathcal{G}'$, then similar to the previous proof, the difference in the scores $\mathrm{Score}(\mathcal{G}') - \mathrm{Score}(\mathcal{G})$ can be computed as follows:

$$\mathrm{Score}(\mathcal{G}', \hat{X}) - \mathrm{Score}(\mathcal{G}, \hat{X}) = -\frac{1}{2}\sum_{k \in \mathcal{I}_{j \setminus i}} \log\left(1 - (\hat{\rho}_{i,j|S}^k)^2\right) + \frac{1}{2}\sum_{k \in \mathcal{I}_{i \setminus j}} \log\left(1 - (\hat{\rho}_{i,j|S}^k)^2\right)$$

Since $n_1 = Cn_2 = Cn_3$ and there is no arrow between $4$ and $5$ in either of $\mathcal{G}^2$ or $\mathcal{G}^3$, then the distributions of $\hat{\rho}_{4,5|S}^{\mathcal{I}-5}$ and $\hat{\rho}_{4,5|S}^{\mathcal{I}-4}$ are identical. Therefore, $\mathrm{Score}(\mathcal{G}', \hat{X}) < \mathrm{Score}(\mathcal{G}, \hat{X})$ with probability $\frac{1}{2}$. $\qquad \square$

## C  Proof of Theorem 4.1

Recall that a DAG $\mathcal{H}$ is called an *independence map* of a DAG $\mathcal{G}$, denoted $\mathcal{G} \leq \mathcal{H}$, if every CI relation entailed by the $d$-separation statements of $\mathcal{H}$ are also entailed by $\mathcal{G}$. The proof of Theorem 4.1 relies on the transformational relationship between a DAG and an independence map given in [1, Theorem 4]. In short, the theorem states that for an independence map $\mathcal{G} \leq \mathcal{H}$, there exists a sequence of covered arrow reversals and arrow additions such that after each arrow reversal or addition, the resulting DAG $\mathcal{G}'$ satisfies $\mathcal{G} \leq \mathcal{G}' \leq \mathcal{H}$, and after all arrow reversals and additions $\mathcal{G}' = \mathcal{H}$. The proof of this fact follows from the APPLY-EDGE OPERATION algorithm [1], which describes the choices that can be made to produce such a transformation of independence maps. In [7] the authors refer to the sequence of independence maps

$$\mathcal{G} \leq \mathcal{G}^{(1)} \leq \mathcal{G}^{(2)} \leq \cdots \leq \mathcal{G}^{(m-1)} \leq \mathcal{H}$$

that transforms $\mathcal{G}$ into $\mathcal{H}$ as a *Chickering sequence*.

A key feature of the APPLY-EDGE OPERATION algorithm is that it recurses on the common sink nodes between $\mathcal{G}$ and $\mathcal{H}$. Namely, if $\mathcal{G}$ and $\mathcal{H}$ have any sink nodes with the same set of parents in both DAGs, the algorithm first deletes these nodes and compares the resulting subDAGs. This is repeated until there are no sink nodes in the two graphs with the exact same set of parents. The remaining set of sink nodes that must be fixed are denoted $s_1 \ldots, s_M$. Then the algorithm begins to reverse and add arrows to the relevant subDAG of $\mathcal{G}$ until a new common sink node appears, which it then

deletes, and so on. Once the algorithm corrects one such sink node in the subDAG of $\mathcal{G}$ to match the same node in the subDAG of $\mathcal{H}$, we say the sink node has been *resolved*. In [7] the authors observe that if we have an independence map of minimal I-MAPs $\mathcal{G}_\pi \leq \mathcal{G}_\tau$, then there exists a Chickering sequence that adds arrows and reverses arrows so that exactly one sink node is resolved at a time; i.e., there is no need to do arrow reversals and additions to any one sink node in order to resolve another. To prove Theorem 4.1, we utilize this fact and the following two lemmas.

**Lemma C.1.** *Suppose $\mathcal{G}$ is an independence map of the data-generating DAG $\mathcal{G}_{\pi^*}$ for the permutation $\pi$. Let $i \rightarrow j$ denote a covered edge in $\mathcal{G}$, and let $S$ denote the set of nodes that precedes $i$ in permutation $\pi$; i.e.,*

$$S := \{\ell \mid \pi(\ell) < \pi(i)\},$$

*then in $\mathcal{G}_{\pi^*}$ the set of d-connecting paths from $i$ to $j$ given $S$ is the same as the set of d-connecting paths from $i$ to $j$ given $\mathrm{Pa}_{\mathcal{G}}(i)$.*

*Proof.* We prove this by contradiction. Suppose in $\mathcal{G}_{\pi^*}$ there exists a path $P_{i \rightarrow j}$ that d-connects $i$ and $j$ given $S$ but $P_{i \rightarrow j}$ is d-separated given $\mathrm{Pa}_{\mathcal{G}}(i)$. Then there must exist at least one node $a \in S \setminus \mathrm{Pa}_{\mathcal{G}}(i)$ that is a collider on $P_{i \rightarrow j}$ or a descendent of a collider on $P_{i \rightarrow j}$. If $a$ is a collider on $P_{i \rightarrow j}$, then $a$ d-connects $i$ given $S \setminus \{a\}$. If no such collider exists, then $a$ must be a descendent of a collider $s$ on $P_{i \rightarrow j}$. Moreover, $a$ d-connects $s$ given $S \setminus \{a\}$ and $s$ also d-connects $i$ given $S \setminus \{a\}$. Since $s \notin S$, $a$ d-connects $i$ given $S \setminus \{a\}$. However, since $\mathcal{G}$ is an independence map, $a$ must be a parent of node $i$ in $\mathcal{G}$, which contradicts with the fact that $a \notin \mathrm{Pa}_{\mathcal{G}}(i)$.

Suppose in $\mathcal{G}_{\pi^*}$ there exists a path $P_{i \rightarrow j}$ that $d$-connects $i$ and $j$ given $\mathrm{Pa}_{\mathcal{G}}(i)$ but is $d$-separated given $S$. Then there must exist some nodes in $S \setminus \mathrm{Pa}_{\mathcal{G}}(i)$ that are non-colliders on $P_{i \rightarrow j}$. Let $a$ denote one of such nodes that is closest to $i$ on $P_{i \rightarrow j}$, then $a$ and $i$ must be d-connected given $S \setminus \{a\}$. Since $\mathcal{G}$ is an independence map, $a$ must be a parent of node $i$ in $\mathcal{G}$, which contradicts with the fact that $a \notin \mathrm{Pa}_{\mathcal{G}}(i)$. $\qquad\square$

**Lemma C.2.** *Given a permutation $\pi$ consider the sequence of minimal I-MAPs from $\mathcal{G}_\pi$ to the data-generating DAG $\mathcal{G}_{\pi^*}$ given by covered arrow reversals*

$$\mathcal{G}_\pi = \mathcal{G}_{\pi^0} \geq \mathcal{G}_{\pi^1} \geq \cdots \geq \mathcal{G}_{\pi^M} = \mathcal{G}_{\pi^*}.$$

*If the edge $i \rightarrow j$ is reversed in $\mathcal{G}_{\pi^{k-1}}$ to produce $\mathcal{G}_{\pi^k}$, then in $\mathcal{G}_{\pi^*}$ all d-connecting paths from $j$ to $i$ given $\mathrm{Pa}_{\mathcal{G}_{\pi^{k-1}}}(i)$ must be pointing towards $i$ (i.e. the edge incident to $i$ on the path points to $i$).*

*Proof.* By [7, Theorem 15], we know there exists a Chickering sequence from $\mathcal{G}_{\pi^*}$ to $\mathcal{G}_\pi$

$$\mathcal{G}_{\pi^*} = \mathcal{G}^0 \leq \mathcal{G}^1 \leq \cdots \leq \mathcal{G}^N = \mathcal{G}_\pi$$

that resolves one sink at a time and, respectfully, reverses one edge at a time. Let $s_1, \ldots, s_M$ denote the list of sink nodes resolved in the Chickering sequence, labeled so that $s_j$ is the $j^{th}$ sink node resolved in the sequence. More specifically, this means that the Chickering sequence can be divided in terms of a sublist of DAGs $\mathcal{G}^{i_1}, \cdots, \mathcal{G}^{i_M}$ such that $\mathcal{G}^{i_j}$ is the DAG produced by resolving sink $s_j$. It follows that the DAGs in the subsequence

$$\mathcal{G}^{i_{j-1}+1} \leq \cdots \leq \mathcal{G}^{i_j - 1}$$

correspond to the arrow additions and covered arrow reversals that are needed to resolve sink $s_j$. For $t = 1, \ldots, q_j$ let $z_t$ denote the node such that $s_j \rightarrow z_t$ must be reversed in order to produce $\mathcal{G}^{i_j}$ from $\mathcal{G}^{i_j-1}$. We label these nodes such that $s_j \rightarrow z_t$ is reversed before $s_j \rightarrow z_{t+1}$ in the given Chickering sequence. Let $\mathcal{G}^{i_j,t}$ denote the DAG generated after reversing edge $s_j \rightarrow z_t$ Then we can write our sequence $G^{i_j-1} \leq \cdots \leq \mathcal{G}^{i_j}$ as:

$$G^{i_j-1} \leq \mathcal{G}^{i_{j-1}+1} \leq \cdots \leq \mathcal{G}^{i_j,t} \leq \cdots \leq \mathcal{G}^{i_j,t+1} \leq \cdots \leq \mathcal{G}^{i_j,q_j-1} \leq \mathcal{G}^{i_j,q_j} = \mathcal{G}^{i_j}.$$

To prove the lemma, we must then show that for all $j$ and $t$, all $d$-connecting paths in $\mathcal{G}_{\pi^*}$ from $s_j$ to $z_t$ given $\mathrm{Pa}_{\mathcal{G}^{i_j,t}}(z_t)$ are pointing towards $z_t$.

To prove this, let $\pi^{j-1}$ denote a permutation consistent with $\mathcal{G}^{i_j-1}$ and let $S_{\pi^{j-1}}(z_t)$ denote the set of nodes that precedes $z_t$ in the permutation $\pi^{j-1}$; i.e.,

$$S_{\pi^{j-1}}(z_t) := \{\ell \mid \pi^{j-1}(\ell) < \pi^{j-1}(z_t))\}.$$

If $\pi^{j-1} = \ldots s_j \ldots z_1 \ldots z_t \ldots z_{q_j} \ldots$, then for $t = 1, \ldots, q_j$, we can always choose a linear extension $\pi^{j,t}$ of $\mathcal{G}^{i_{j,t}}$ in which $\pi^{j,t} = \ldots z_1 \ldots z_t s_j \ldots z_{q_j} \ldots$, and $S_{\pi^{j-1}}(z_t) \backslash \{s_j\} = S_{\pi^{j,t}}(z_t)$ by moving $s_j$ forward in the permutation $\pi^{j-1}$ until it directly follows $z_t$. It is always possible to pick such an extension as $\pi^{j,t}$ since we can choose the extension of $\pi^{j-1}$ so that the only nodes in between $z_{t-1}$ and $z_t$ are the descendants of $z_{t-1}$ that are also ancestors of $z_t$. The existence of such an ordering of $\pi$ with respect to the ordering of the nodes $z_1, \ldots, z_{q_j}$ is implied by the choice of the maximal child in each iteration of step 5 of the APPLY-EDGE OPERATION algorithm that produces the Chickering sequence [1]. Using Lemma C.1, we know that any $d$-connecting path from $z_t$ to $s_j$ given $S_{\pi^{j,t}}(z_t)$ in $\mathcal{G}_{\pi^*}$ is actually the same as a $d$-connecting path from $z_t$ to $s_j$ in $\mathcal{G}_{\pi^*}$ given $\mathrm{Pa}_{\mathcal{G}^{i_{j,t}}}(z_t)$. Since $S_{\pi^{j,t}}(z_t) = S_{\pi^{j-1}}(z_t) \backslash \{s_j\}$ then it remains to show that any $d$-connecting path from $s_j$ to $z_t$ given $S_{\pi^{j-1}}(z_t) \backslash \{s_j\}$ in $\mathcal{G}_{\pi^*}$ goes to $z_t$. Since $s_j \to z_t$ in $\mathcal{G}_{\pi^{j-1}}$, we prove the following, slightly stronger, statement: for any edge $a \to b \in \tilde{\mathcal{G}}^{i_{j-1}}$, all d-connecting paths from $a$ to $b$ given $S_{\pi^{j-1}}(b) \setminus \{a\}$ in $\mathcal{G}_{\pi^*}$ go to $b$.

We prove this stronger statement via induction. If $\tilde{\mathcal{G}}^{i_{j-1}} = \tilde{\mathcal{G}}_{\pi^*}$, the statement is definitely true since the only possible $d$-connection between $a$ and $b$ given $S_{\pi^*}(b) \setminus \{a\}$ is the arrow $a \to b \in \mathcal{G}_{\pi^*}$. Suppose it is also true when $j = j' - 1$. Recall the only difference between $\pi^{j'-1}$ and $\pi^{j'}$ is the position of $s_{j'}$. If in $\tilde{\mathcal{G}}^{i_{j'}}$ there is a new arrow $a \to b$, then this arrow corresponds to some paths that $d$-connect $a$ and $b$ given $S_{\pi^{j'}}(b) \setminus \{a\}$. However, they are $d$-separated given $S_{\pi^{j'-1}}(b) \setminus \{a\}$. Since $S_{\pi^{j'}}(b) = S_{\pi^{j'-1}}(b) \setminus \{s_{j'}\}$, then $s_{j'}$ must be in the middle of these new paths and is not a collider. In this case, removing $s_{j'}$ from the conditioning set would turn these paths into $d$-connections.

Without loss of generality, we consider one of these new paths from $a$ to $b$ denoted as $P_{a \to b}$. Since $s_{j'}$ is in the middle of $P_{a \to b}$, let $P_{s_{j'} \to b}$ denote the latter part of $P_{a \to b}$. Obviously, $P_{s_{j'} \to b}$ also $d$-connects $s_{j'}$ and $b$ given $S_{\pi^{j'-1}}(b) \setminus \{s_{j'}\}$. As $\mathcal{G}^{i_{j'-1}}$ is an independence map of $\mathcal{G}_{\pi^*}$, $s_{j'} \to b$ must be an edge in $\mathcal{G}^{i_{j'-1}}$, and therefore it also exists in $\tilde{\mathcal{G}}^{i_{j'-1}}$. Notice, if $s_{j'} \to b \in \tilde{\mathcal{G}}^{i_{j'-1}}$ then, in $\mathcal{G}_{\pi^*}$, all paths that $d$-connect $s_{j'}$ and $b$ given $S_{\pi^{j'-1}}(b) \setminus \{s_{j'}\}$ go to $b$. Therefore, $P_{s_j \to b}$ is a path that goes to $b$. In this case, $P_{a \to b}$ is also a path that goes to $b$. As there is no specification of $P_{a \to b}$, this holds for all new paths, and this completes the proof. $\square$

*Proof of Theorem 4.1.* We can now prove Theorem 4.1. Let $\mathbb{P}$ be a distribution that is faithful with respect to an unknown I-MAP $\mathcal{G}_{\pi^*}$. Suppose that observational and interventional data are drawn from $\mathbb{P}$ for a collection of interventional targets $\mathcal{I} = \{I_1 := \emptyset, I_2, \ldots, I_K\}$, and that $\mathbb{P}^k$ is faithful to $\mathcal{G}_{\pi^*}^k$ for all $k \in [K]$. We must show that Algorithm 1 returns to $\mathcal{I}$-MEC of $\mathcal{G}_{\pi^*}$. Suppose that we are at the DAG $\mathcal{G}_\pi$ for some permutation $\pi$ of $[p]$. By [7, Theorem 15] there exists a sequence of minimal I-MAPS

$$\mathcal{G}_{\pi^*} = \mathcal{G}_{\pi^M} \leq \mathcal{G}_{\pi^{M-1}} \leq \cdots \leq \mathcal{G}_{\pi^0} = \mathcal{G}_\pi,$$

where $\mathcal{G}_{\pi^k}$ is produced from $\mathcal{G}_{\pi^{k-1}}$ by reversing a covered arrow $i \to j$ and then deleting some edges of $\mathcal{G}_{\pi^{k-1}}$. In particular, this sequence arises from a Chickering sequence that resolves one sink node at a time, as in Lemma C.2. We would now like to see that for such a path

$$\mathrm{Score}(\mathcal{G}_{\pi^k}) \geq \mathrm{Score}(\mathcal{G}_{\pi^{k-1}}) - \delta_n,$$

for all $k = 1, 2, \ldots, M$. Suppose first that $\mathcal{G}_{\pi^{k-1}}$ and $\mathcal{G}_{\pi^k}$ differ only by a covered arrow reversal (i.e., they have the same skeleton). Using the notation from the previous proofs, we let $\hat{\rho}_{i,j|S}^k$ denote the value of the partial correlation of $i, j \mid S$ for some set $S \subset [p]$ based on data $\hat{X}^k$ from the intervention $I_k$. If we take $S = \mathrm{Pa}_i(\mathcal{G}_{\pi^{k-1}})$, then by [5, Lemma 5.1] and Lemma C.2 it follows that

$$\mathrm{Score}(\mathcal{G}_{\pi^k}) - \mathrm{Score}(\mathcal{G}_{\pi^{k-1}}) = -\frac{1}{2} \sum_{k \in \mathcal{I}_{j \backslash i}} \left( \log \left( 1 - (\hat{\rho}_{i,j|S}^k)^2 \right) + \lambda_{n_k} \right)$$

$$+ \frac{1}{2} \sum_{k \in \mathcal{I}_{i \backslash j}} \left( \log \left( 1 - (\hat{\rho}_{i,j|S}^k)^2 \right) + \lambda_{n_k} \right).$$

Note that the value of $\sum_{k \in \mathcal{I}_{i \backslash j}} \left( \log \left( 1 - (\hat{\rho}_{i,j|S}^k)^2 \right) + \lambda_{n_k} \right)$ will be zero when the set $\mathcal{I}_{i \backslash j}$ is empty. It then follows from Lemma C.2 that $\mathrm{Score}(\mathcal{G}_{\pi^k}) \geq \mathrm{Score}(\mathcal{G}_{\pi^{k-1}}) - \delta_n$, for all $k = 1, \ldots, M$.

The above argument shows that if two minimal I-maps $\mathcal{G}_{\pi^k}$ and $\mathcal{G}_{\pi^{k-1}}$ along the given sequence are in the same MEC then their relative scores in Algorithm 1 are at most nondecreasing. Thus, it only remains to show that if $\mathcal{G}_{\pi^{k-1}}$ is not in the $\mathcal{I}$-MEC of $\mathcal{G}_{\pi^*}$ then

$$\text{Score}(\mathcal{G}_{\pi^*}) > \text{Score}(\mathcal{G}_{\pi^{k-1}}).$$

Since $\mathcal{G}_{\pi^{k-1}}$ and $\mathcal{G}_{\pi^*}$ are not $\mathcal{I}$-Markov equivalent then, by [2, Theorem 10], there is at least one $I_t \in \mathcal{I}$ such that $\mathcal{G}_{\pi^{k-1}}^t$ and $\mathcal{G}_{\pi^*}^t$ have different skeletons. However, in this case $\text{Score}(\mathcal{G}_{\pi^*}) > \text{Score}(\mathcal{G}_{\pi^{k-1}})$ since the interventional distribution $\mathbb{P}^t$ is faithful to the DAG $\mathcal{G}_{\pi^*}^t$. $\qquad\square$

# D   Proof of Theorem 4.4

We would now like to prove that Algorithm 2 is consistent under the faithfulness assumption. That is, suppose we are given a collection of interventional targets $\mathcal{I} = \{I_1 = \emptyset, I_2, \ldots, I_K\}$ and data drawn from the distributions $\mathbb{P}^1, \ldots, \mathbb{P}^K$, all of which are faithful to the (respective) interventional DAGs $\mathcal{G}_{\pi^*}^1, \ldots, \mathcal{G}_{\pi^*}^K$. Then Algorithm 2 will return a DAG that is $\mathcal{I}$-Markov equivalent to $\mathcal{G}_{\pi^*}$. In [7], the authors show that there exists a sequence of I-MAPs given by covered arrow reversals

$$\mathcal{G}_\pi \geq \mathcal{G}_{\pi^1} \geq \cdots \geq \mathcal{G}_{\pi^{m-1}} \geq \mathcal{G}_{\pi^m} \geq \cdots \geq \mathcal{G}_{\pi^M} \geq \mathcal{G}_{\pi^*}$$

taking us from any $\mathcal{G}_\pi$ to the data-generating DAG $\mathcal{G}_{\pi^*}$. We must now show that there exists such a sequence using only $\mathcal{I}$-covered arrow reversals.

**Theorem D.1.** *For any permutation $\pi$, there exists a list of $\mathcal{I}$-covered arrow reversals from $\mathcal{G}_\pi$ to the data-generating DAG $\mathcal{G}_{\pi^*}$*

$$\mathcal{G}_\pi = \mathcal{G}_{\pi^0} \geq \mathcal{G}_{\pi^1} \geq \cdots \geq \mathcal{G}_{\pi^{m-1}} \geq \mathcal{G}_{\pi^m} \geq \cdots \geq \mathcal{G}_{\pi^{M-1}} \geq \mathcal{G}_{\pi^M} = \mathcal{G}_{\pi^*}$$

*Proof.* Suppose that $\mathcal{G}_{\pi^m}$ is produced from $\mathcal{G}_{\pi^{m-1}}$ via reversing the covered arrow $i \to j$ in $\mathcal{G}_{\pi^{m-1}}$ and let $S = \text{Pa}_{\mathcal{G}_{\pi^{m-1}}}(i)$. By Lemma C.2, it must be that $i$ and $j$ are $d$-connected in $\mathcal{G}_{\pi^*}$ given $S$ only by paths for which the arrow incident to $i$ points towards $i$. It follows that for $k \in \mathcal{I}_{i\setminus j}$ there are no paths $d$-connecting $i$ and $j$ in $\mathcal{G}_{\pi^*}^k$. Therefore, $i \to j \notin \mathcal{G}_{\pi^{m-1}}$ for all $k \in \mathcal{I}_{i\setminus j}$; i.e., the arrow $i \to j$ is $\mathcal{I}$-covered in $\mathcal{G}_{\pi^{m-1}}$. $\qquad\square$

The previous theorem states that we can use only $\mathcal{I}$-covered arrow reversals to produce a sequence of I-MAPs taking us from any DAG $\mathcal{G}_\pi$ to the data-generating DAG $\mathcal{G}_{\pi^*}$. In the case that $\mathcal{G}_{\pi^{m-1}}$ and $\mathcal{G}_{\pi^m}$ are in different MECs it follows from the construction of such a sequence of minimal I-MAPs under the faithfulness assumption in the observational setting that $\mathcal{G}_{\pi^{m-1}}$ has strictly more arrows than $\mathcal{G}_{\pi^m}$. It remains to show that if $\mathcal{G}_{\pi^{m-1}}$ and $\mathcal{G}_{\pi^m}$ are in the true MEC then $\mathcal{G}_{\pi^{m-1}}$ has strictly more $\mathcal{I}$-contradicting arrows than $\mathcal{G}_{\pi^m}$ whenever they are not in the same $\mathcal{I}$-MEC and they have exactly the same $\mathcal{I}$-contradicting arrows when in the same $\mathcal{I}$-MEC. This is the content of the following theorem.

**Theorem D.2.** *Suppose that the distributions $\mathbb{P}^1, \ldots, \mathbb{P}^K$ are faithful to their respective interventional DAGs $\mathcal{G}_{\pi^*}^1, \ldots, \mathcal{G}_{\pi^*}^K$. For any permutation $\pi$ such that $\mathcal{G}_\pi$ and $\mathcal{G}_{\pi^*}$ are in the same MEC there exists a list of $\mathcal{I}$-covered arrow reversals from $\mathcal{G}_\pi$ to $\mathcal{G}_{\pi^*}$*

$$\mathcal{G}_\pi = \mathcal{G}_{\pi^0} \geq \mathcal{G}_{\pi^1} \geq \cdots \geq \mathcal{G}_{\pi^{m-1}} \geq \mathcal{G}_{\pi^m} \geq \cdots \geq \mathcal{G}_{\pi^{M-1}} \geq \mathcal{G}_{\pi^M} = \mathcal{G}_{\pi^*}$$

*such that, for all $m \in [M]$, either $\mathcal{G}_{\pi^{m-1}}$ and $\mathcal{G}_{\pi^m}$ are in the same $\mathcal{I}$-MEC or $\mathcal{G}_{\pi^m}$ is produced from $\mathcal{G}_{\pi^{m-1}}$ by the reversal of an $\mathcal{I}$-contradicting arrow. Moreover, the number of $\mathcal{I}$-contradicting arrows in $\mathcal{G}_{\pi^m}$ is strictly less than the number of $\mathcal{I}$-contradicting arrows in $\mathcal{G}_{\pi^{m-1}}$.*

*Proof.* Suppose that $\mathcal{G}_{\pi^m}$ is produced from $\mathcal{G}_{\pi^{m-1}}$ by reversing the $\mathcal{I}$-covered arrow $i \to j$ in $\mathcal{G}_{\pi^{m-1}}$. If $\mathcal{I}_{i\setminus j} = \mathcal{I}_{j\setminus i} = \emptyset$ then $\mathcal{G}_{\pi^{m-1}}$ and $\mathcal{G}_{\pi^m}$ are in the same $\mathcal{I}$-MEC and hence $i \to j$ is not an $\mathcal{I}$-contradicting arrow.

Otherwise, they must belong to different $\mathcal{I}$-MECs and we have that $\mathcal{I}_{i\setminus j} \cup \mathcal{I}_{j\setminus i} \neq \emptyset$. Let $S = \text{Pa}_{\mathcal{G}_{\pi^{m-1}}}(i)$, by Lemma C.2. It must be that $i$ and $j$ are $d$-connected in $\mathcal{G}_{\pi^*}$ given $S$ only by paths for which the arrow incident to $i$ points towards $i$. Since $\mathcal{G}_{\pi^{m-1}}$ is in the true MEC then $i$ and $j$ must be adjacent in $\mathcal{G}_{\pi^*}$. It then follows from Lemma C.2 that the arrow between $i$ and $j$ points

to $i$. In other words, $j \rightarrow i \in \mathcal{G}_{\pi^*}$. In this case, for all $k \in \mathcal{I}_{j \setminus i}$ we have, under the faithfulness assumption, that $\mathbb{P}^k$ satisfies $i \not\perp\!\!\!\perp j$ since the arrow $j \rightarrow i$ in $\mathcal{G}_{\pi^*}$ is not deleted in the interventional DAG $(\mathcal{G}_{\pi^*})^k$. Similarly, for all $k \in \mathcal{I}_{i \setminus j}$ we know $i \perp\!\!\!\perp j$ in $\mathbb{P}^k$ since all $d$-connecting paths between $i$ and $j$ in the interventional DAG $(\mathcal{G}_{\pi^*})^k$ must be given by conditioning on some descendants of $i$. Thus, $i$ and $j$ are $d$-separated given $\emptyset$ in $(\mathcal{G}_{\pi^*})^k$, and it follows from the Markov assumption that $i \perp\!\!\!\perp j$ in $(\mathcal{G}_{\pi^*})^k$. Therefore, by Definitions 4.2 and 4.3, we know that $i \rightarrow j$ is an $\mathcal{I}$-covered arrow that is also $\mathcal{I}$-contradicting. Furthermore, since the reversal of an $\mathcal{I}$-contradicting arrow makes it not $\mathcal{I}$-contradicting and the $\mathcal{I}$-contradicting arrows of $\mathcal{G}_{\pi^m}$ are contained in the $\mathcal{I}$-contradicting arrows of $\mathcal{G}_{\pi^{m-1}}$, it follows that $\mathcal{G}_{\pi^m}$ has strictly less $\mathcal{I}$-contradicting arrows than $\mathcal{G}_{\pi^{m-1}}$. $\qquad\square$

*Proof of Theorem 4.4* The proof of this theorem follows immediately from Theorem D.1 and Theorem D.2. This is because Theorem D.1 implies that under the faithfulness assumption there is a sequence of $\mathcal{I}$-covered arrow reversals by which we can reach the true MEC, and Theorem D.2 implies that we then use $\mathcal{I}$-contradicting arrows to reach the true $\mathcal{I}$-MEC within the true MEC. Moreover, Theorem D.2 implies that the true $\mathcal{I}$-MEC will contain DAGs with the fewest $\mathcal{I}$-contradicting arrows. $\qquad\square$

# E   Supplementary Material for Real Data Analysis

This section contains supplementary information about the real data analysis. Table E.1 and Figure E.1 present additional details about the perturb-seq experiments. Table E.2 shows more details about the flow cytometry dataset. Table E.3 compares the results of IGSP and k-IGSP with other methods that allow latent confounders as applied to the Sachs et al. [6] dataset. Figures E.2 and E.3 are our reconstructions of the causal gene network for the perturb-seq data set and the protein signaling network for the Sachs et al. dataset, respectively.

Figure E.1: Heatmap of the true effects of each gene deletion on each measured gene. All 56 guide RNAs used in the experiment are listed on the x-axis and measured genes are listed on the y-axis. 18 of 56 guides, which target 8 genes in total, were selected for analysis because they were effective. Red (positive on q-value scale) indicate gene deletions that increase abundance of the measured gene. Blue (negative on q-value scale) indicate gene deletions that decrease abundance of the measured gene. White (zero on q-value scale) indicates no observed effect of gene deletion.

Table E.1: Number of samples under each gene deletion for processed perturb-seq dataset

| Intervention: | None | Stat2 | Stat1 | Rel | Hif1a | Spi1 | Nfkb1 | Rela | Cebpb |
|---|---|---|---|---|---|---|---|---|---|
| # Samples: | 992 | 2426 | 3337 | 1513 | 301 | 796 | 3602 | 1068 | 392 |

Table E.2: Number of samples under each protein intervention for flow cytometry dataset

| Intervention: | None | Akt | PKC | PIP2 | Mek | PIP3 |
|---|---|---|---|---|---|---|
| # Samples: | 1755 | 911 | 723 | 810 | 799 | 848 |

Table E.3: Interaction prediction results of IGSP and k-IGSP and other methods that allow for latent variables. Here the consensus network from [6] is denoted by [6]a and their reconstructed network by [6]b. For [4] we provide results from both ICP and hidden ICP, denoted by [4]a and [4]b respectively. For IGSP and k-IGSP, we chose the standardly used significance level $\alpha = 0.05$ as the cut-off for CI testing, which resulted in a similar number of predicted interactions as in [4].

| Edge | [6]a | [6]b | [4]a | [4]b | ACI [3] | COmbINE [8] | IGSP | k-IGSP |
|---|---|---|---|---|---|---|---|---|
| RAF → MEK | ✓ | ✓ | | | ✓ | ✓ | | |
| RAF → JNK | | | | | | | | ✓ |
| MEK → RAF | | | ✓ | ✓ | | ✓ | ✓ | ✓ |
| MEK → ERK | ✓ | ✓ | | | ✓ | | | |
| MEK → AKT | | | | | ✓ | | | |
| MEK → JNK | | | | | ✓ | | | |
| PLCg → PIP2 | ✓ | ✓ | ✓ | ✓ | | ✓ | | |
| PLCg → PIP3 | | ✓ | | | | ✓ | ✓ | |
| PLCg → PKC | ✓ | | | | | | | |
| PIP2 → PLCg | | | ✓ | | ✓ | ✓ | | ✓ |
| PIP2 → PIP3 | | | | | | | | ✓ |
| PIP2 → PKC | ✓ | | | | | | | |
| PIP3 → PLCg | ✓ | | | | | ✓ | | ✓ |
| PIP3 → PIP2 | ✓ | ✓ | ✓ | ✓ | | ✓ | ✓ | |
| PIP3 → AKT | ✓ | | | | | | | |
| AKT → ERK | | | ✓ | ✓ | | ✓ | ✓ | ✓ |
| AKT → PKA | | | | | | ✓ | ✓ | |
| AKT → JNK | | | | | ✓ | | | |
| ERK → AKT | | ✓ | ✓ | ✓ | | ✓ | | |
| PKA → RAF | ✓ | ✓ | | | ✓ | | | |
| PKA → MEK | ✓ | ✓ | | ✓ | ✓ | | | |
| PKA → ERK | ✓ | ✓ | ✓ | | ✓ | ✓ | | ✓ |
| PKA → AKT | ✓ | ✓ | | ✓ | ✓ | | | ✓ |
| PKA → PKC | | | | | | | | |
| PKA → P38 | ✓ | ✓ | | | ✓ | | | |
| PKA → JNK | ✓ | ✓ | | | ✓ | | | |
| PKC → RAF | ✓ | ✓ | | | ✓ | | | |
| PKC → MEK | ✓ | ✓ | | | ✓ | | | |
| PKC → PLCg | | | | | ✓ | | | |
| PKC → PIP2 | | | | | ✓ | | | |
| PKC → PIP3 | | | | | ✓ | | | |
| PKC → ERK | | | | | ✓ | | | |
| PKC → AKT | | | | | ✓ | | | |
| PKC → PKA | | ✓ | | | | | | |
| PKC → P38 | ✓ | ✓ | | ✓ | ✓ | | ✓ | ✓ |
| PKC → JNK | ✓ | ✓ | ✓ | ✓ | ✓ | | ✓ | ✓ |
| P38 → JNK | | | | ✓ | | ✓ | | ✓ |
| P38 → PKC | | | | | ✓ | | | |
| JNK → PKC | | | | | ✓ | | | |
| JNK → P38 | | | | | ✓ | ✓ | | |