[Reviews · NeurIPS 2017]

Reviewer 1



Update after the authors' response. Thanks for your answer. Very happy about the scientific discussion, this is a really nice piece of work, and you convinced me you know the subject very well and your contributions are now much clearer to me. I really encourage you to consider all the points raised by all reviewers (reading the reviews by others also helped) to make the best possible paper in the end! Good luck with your paper. ################################# This work is anchored in a Bayesian network (BN) framework for causal inference from interventional and observational data. The authors built from existing works in several directions: 2 algorithms are presented (one showing consistency only under conditions which are difficult to control) with their respective theoretical guarantees. Simulated and real world data are used to demonstrate the method performance. Remarks: - l18: careful, all BN are not causal; distribution-based BN have a different interpretation of their edges in terms of (in)dependence relationships. I know you are not fooled, bu try making it clearer. - section 2: you missed at least Rau et al. BMC System Biol 2013. - the covered arrow concept is not 100% clear to me, can you give a little bit more simple interpretation on its meaning if possible? Usefulness? - l91: intervention DAG, why don't you simply say that it consists of the initial DAG without edges pointing at the nodes which was intervened? - l115: interesting example. Perhaps explaining it in details in the paper only once (it is back on p4 avfter Th4.1) would be a very good idea. - Greedy SP essentially seeks an ordering of the variables. If the ordering is fixed, the problem is 'easy'. If not, an adequate way to browse graph configurations is needed. E.g. see https://arxiv.org/abs/1507.02018 - Eqn after l132 looks like a global structural Lasso Equation. Can you comment a bit please? - Th4.1: is the result 'almost surely'? As nj's \to \infty? - l165: skeleta? - simulations seem a bit too simple and not very high-dimensional...when the real data is complex and high-dimensional?!

Reviewer 2



In this paper the authors propose two permutation based algorithms for causal structure learning from a mix of observational and interventional data. The authors assume that their data does not contain hidden variables and that the interventional data comes from a do-type intervention, where it is known which variables were intervened on. Their second algorithm, called IGSP, is fully non-parametric and relies primarily on conditional independence tests. Previous work on this topic is the GIES algorithm from Hauser and Buhlmann (2012). The authors show that GIES is in fact inconsistent, whereas IGSP is consistent under the faithfulness assumption. Additionally, for a linear Gaussian setting, the authors show that IGSP and GIES perform comparably, whereas it is expected that IGSP performs better in non-linear or non-Gaussian settings as it is non-parametric. This is very nice and comprehensive work on a very important topic. Since their IGSP algorithm is non-parametric and relies on conditional independence tests, I was wondering if the authors consider extending it to the hidden variable case? As for specific comments on the paper: It would be beneficial to make the implementation of the proposed algorithms available. Additionally, it would be interesting to know the computational complexity of IGSP or at least empirically compare the runtime of IGSP and GIES. As for the example in analysis of protein signaling data: You use the data from Sachs et al. (2005) which is purely interventional, but it is my understanding that your method specifically assumes it also has access to observational data? Do you do anything to remedy that, or is this just an example where the model assumptions are violated? In line 204: \alpha_{ij} should be a_{ij}

Reviewer 3



The paper describes two permutation-based causal inference algorithms that can deal with interventional data. These algorithms extend a recent hybrid causal inference algorithm, Greedy SP, which performs a greedy search on the intervention equivalence classes, similar to GIES [8], but based on permutations. Both of the proposed algorithms are shown to be consistent under faithfulness, while GIES is shown not to be consistent in general. Algorithm1 is computationally inefficient, so the evaluation is performed only with Algorithm 2 (IGSP) with two different types of independence criteria (linear Gaussian and kernel based). The evaluation on simulated data shows that IGSP is more consistent than GIES. Moreover, the authors apply the algorithm also to two biological datasets, a well-known benchmark dataset (Sachs et al.) and a recent dataset with gene expressions. Quality Pros: - I didn’t check in depth the proofs, but to the best of my knowledge the paper seems to be technically sound. - It seems to provide a good theoretical analysis in the form of consistency guarantees, as well as an empirical evaluation on both simulated and real-world datasets. - The evaluation on the perturb-seq seems to be methodologically quite good. Cons: - The paper doesn’t seem to discuss a possibly obvious drawback: the potentially higher order conditional independence tests may be very brittle in the finite data case. - The evaluation compares only with GIES, while there are other related methods, e.g. some of the ones mentioned below. - The empirical evaluation on real-world data is not very convincing, although reconstructing causal networks in these settings reliably is known to be very difficult. - For the Sachs data, the paper seems to suggest that the consensus network proposed in the original paper is the true causal graph in the data, while it is not clear if this is the case, so I think Figure 4 is a bit misleading. Many methods have been evaluated on this dataset, so the authors could include some more comparisons, at least in the Supplement. Clarity Pros: - The paper is well-organized and clearly written. - It is quite clear how the approach is different from previous work. - From a superficial point of view, the proofs in the Appendix seem to be quite well-written, which seems to be quite rare among the other submissions. Cons: - Some parts of the background could be expanded to make it more self-contained, for example the description of Greedy SP. - It probably sounds obvious to the authors, but maybe it would be good to point out in the beginning of the paper that the paper considers the causal sufficient case, and the case in which the intervention target is known. - For the Sachs data, I would possibly use a Table of causal relations similar to the one in the Supplement of [13] to compare also with other methods. Original Pros: - The paper seems to provide an interesting extension of Greedy SP to include interventional datasets, which allows it to be applied on some interesting real-world datasets. Cons: - The related work section could refer to the several constraint-based (most of them in practice hybrid) methods that have been developed recently to reconstruct causal graphs from observational and interventional datasets (even when we cannot assume causal sufficiency), for example: -- A. Hyttinen, F. Eberhardt, and M. Järvisalo: Constraint-based Causal Discovery: Conflict Resolution with Answer Set Programming, Proceedings of the 30th Conference on Uncertainty in Artificial Intelligence, 2014. -- Triantafillou, Sofia, and Ioannis Tsamardinos. "Constraint-based causal discovery from multiple interventions over overlapping variable sets." Journal of Machine Learning Research 16 (2015): 2147-2205. -- Magliacane, Sara, Tom Claassen, and Joris M. Mooij. "Ancestral causal inference." Advances in Neural Information Processing Systems. 2016. - The last paper also offers some (possibly limited) asymptotic consistency guarantees, maybe the authors could discuss its limitations with respect to their guarantees. Significance Pros: - This paper and other permutation-based causal discovery approaches seem to provide a novel and interesting alternative to previous methods Cons: - The applicability of these methods to real-world data is reduced by the assumption that there are no latent confounders. Typos: - References: [13] N. Meinshausen, A. Hauser, J. M. Mooijc, -> Mooij